# Crumpled Graphene-Storage Media for Hydrogen and Metal Nanoclusters

**DOI:** 10.3390/ma14092098

**Published:** 2021-04-21

**Authors:** Liliya R. Safina, Karina A. Krylova, Ramil T. Murzaev, Julia A. Baimova, Radik R. Mulyukov

**Affiliations:** 1Ufa State Petroleum Technological University, Kosmonavtov Str. 1, 450062 Ufa, Russia; radik@imsp.ru; 2Institute for Metals Superplasticity Problems of the Russian Academy of Sciences, Khalturina 39, 450001 Ufa, Russia; bukreevakarina@gmail.com (K.A.K.); mur611@mail.ru (R.T.M.); julia.a.baimova@gmail.com (J.A.B.); 3Bashkir State University, Validy Str. 32, 450076 Ufa, Russia

**Keywords:** crumpled graphene, Ni-graphene composite, hydrogen, molecular dynamics, storage media

## Abstract

Understanding the structural behavior of graphene flake, which is the structural unit of bulk crumpled graphene, is of high importance, especially when it is in contact with the other types of atoms. In the present work, crumpled graphene is considered as storage media for two types of nanoclusters—nickel and hydrogen. Crumpled graphene consists of crumpled graphene flakes bonded by weak van der Waals forces and can be considered an excellent container for different atoms. Molecular dynamics simulation is used to study the behavior of the graphene flake filled with the nickel nanocluster or hydrogen molecules. The simulation results reveal that graphene flake can be considered a perfect container for metal nanocluster since graphene can easily cover it. Hydrogen molecules can be stored on graphene flake at 77 K, however, the amount of hydrogen is low. Thus, additional treatment is required to increase the amount of stored hydrogen. Remarkably, the size dependence of the structural behavior of the graphene flake filled with both nickel and hydrogen atoms is found. The size of the filling cluster should be chosen in comparison with the specific surface area of graphene flake.

## 1. Introduction

The transformation of nanoscale structural elements into three-dimensional (3D) complex architecture is currently an important task of materials science. Since the discovery of fullerenes in 1985, a lot of new carbon structures have been proposed. Carbon polymorphs can be used to obtain nanostructures with unique mechanical and physical properties applicable in nanoelectronics, energy storage devices, sensors, supercapacitors, Li-ion batteries, etc. [1,2,3,4,5,6,7].

Graphene is capable of enhancing the performance, functionality as well as durability of many applications. The one-atom thin structure can serve as the platform for other materials especially since the graphene layer can be bent or crumpled. Extensive studies have been carried out in recent decades to investigate the crumpling behavior of thin sheets like graphene, by both theoretical and experimental methods [8,9,10,11,12,13,14,15,16,17,18,19]. It was shown that crumpled structures can have excellent compression and aggregation-resistant properties [5,6,9,10,20,21,22]. Crumpled graphene can be used for the production of new supercapacitors [19]. However, one of their new and important applications is that of a natural container for other atoms.

It is known that coupling metal nanoclusters with carbon materials can efficiently promote catalysis and electrocatalytic activities [23,24] or prevent the corrosion of metals [25]. redAmong metals, some can be easily attached to its surface, while others even repulse from the surface. The bigger the carbon solubility of metal, the more amounts of carbon can be attracted to the metal surface. Metals such as Ni, Cu, Pt, or Au are the most preferable as catalysts for the growth of graphene [26,27,28]. However, nickel is one of the most broadly used metals for a range of graphene applications, including the growth of carbon nanotubes (CNTs) and graphene [29,30,31]. Lattice mismatch is one of the important factors for choosing the metal for metal–graphene interface and Ni(111) surface is the closest matched interface with respect to the graphene of all transition metals [26]. In [29], a simple and scalable method for the synthesis of hollow graphene balls using a Ni nanocluster template was developed. The interface between graphene and Ni attracts considerable interest due to the possibility of the synthesis of large-area graphene on metal substrate [32,33,34,35,36]. Since the fabrication of graphene layers on the metal substrate was very successful, the idea of the fabrication of metal–graphene layered composites was raised by [37,38,39,40]. Graphene added into the copper and nickel matrix by chemical vapor deposition can considerably improve the strength of metal [38]. The overview of graphene–nickel composites can be found in [39]. Both the theoretical and experimental works on Ni-graphene layered composites showed that the mechanical properties of metal-matrix composite reinforced with the graphene layer considerably improved. In the present work, the other approach to obtain the Ni-graphene composite was applied: introducing metal nanoparticles to the crumpled graphene matrix. In [41,42,43,44] it was shown that such composites can be obtained through compression combined with high temperatures.

The other important advantages of such structures are a large specific surface area (SSA) and, respectively, a high rate of gas adsorption, which makes it possible to predict their use in hydrogen storage [45,46,47,48,49,50,51,52,53]. Various structural parameters can affect the degree of hydrogen accumulation [45]. For example, the amount of adsorbed hydrogen increases with an increase in the CNT diameter, since this increases the surface area on which hydrogen can be adsorbed. An increase in the distance between graphene sheets in the structure also leads to an increase in the gravimetric hydrogen absorption density. Another promising carbon structure is two layers of graphene connected by short CNTs with graphene cones added on top of the structure. The idea of such a structure appeared when it was shown that the hydrogenation of graphene can lead to the buckling of the graphene sheet [54]: the amount of accumulated hydrogen increased from only 3 wt% for the simple graphene plane to 20 wt% for graphene plane with graphene cone. Carbon nanostructures doped with alkali metals (Li and K) can adsorb even more hydrogen [48,49,50,51,52,53]: 14 wt% (Li) and 20 wt% (K) of hydrogen at moderate conditions, in contradiction with the lower values reported later [50]. The Li-doped activated carbon [55] can store from 2.1 to 2.6 wt% of H${}_{2}$ at 77 K and at 2 MPa, which shows that at given pressure–temperature conditions the amount of stored hydrogen can be increased. It should be mentioned that the required amount of stored hydrogen for carbon nanostructures was still not achieved. Thus, an active search for new materials and structures for hydrogen storage and transportation is of high importance.

In the present work, two types of nanofillers for crumpled graphene flake were considered—Ni and hydrogen nanoclusters. Thus, the idea of using crumpled graphene flake for the storage of different atoms using Ni and hydrogen clusters as the example was realized. It is important to understand the dynamics of interaction between single graphene flake and a metal or non-metal nanocluster before the study of storage in three-dimensional crumpled graphene. The behavior of the hybrid structure is studied by atomistic simulation at different temperatures. For the carbon–nickel system, room temperature (300 K) and temperature close to the melting point of Ni (1000 K) were chosen. However, for the carbon–hydrogen system, the temperature of liquid nitrogen (77 K) and room temperature were chosen. The dynamics of the structure were studied during exposure for 20 ps.

## 2. Materials and Methods

To study the interaction between graphene flake (GF) and atomic nanocluster, an initial structure composed of the GF of one size ($N_{C}$ = 252) and atomic clusters of different sizes (*N* = 21; 38; 47; 66; 78) inside the flake was considered (see Figure 1a). One graphene flake was a small CNT(11,11) 1.3 nm-long with two atomic rows deleted along the axis of the nanotube. Since the size of this small sample of graphene along three dimensions is very small (1.3 nm along the *z* axis, 3.6 nm along the *y* axis and one-atom-thick along the *x* axis) it is called “flake” rather than “nanoribbon”. This GF was filled with nanoclusters of two types: (i) nickel nanocluster and (ii) hydrogen nanocluster.

The maximum size of the nanocluster was chosen to almost completely fill the cavity of the graphene flake. The size of the structural elements is shown in Table 1. The distance between the edge of the nanocluster and the side of the flake can be defined as a=(D−d)/2. However, these parameters affect the structural transformation just in the case of metal nanocluster since it is a solid particle. a hydrogen cluster is gas and small H atoms can easily spread inside GF.

It should be mentioned, that nanocluster Ni${}_{78}$ was considered here as nanocluster of critical size for the chosen size of the GF. However, for the metal cluster, this size was too big and the sides of GF were closer than it should have been in a real system. To eliminate the negative effect of the overly sized nanocluster, nanocluster Ni${}_{66}$ was considered the characteristic for the case of metal–graphene interaction. While for the case of hydrogen–graphene interaction, an even bigger size of the nanocluster could be considered since the size of the hydrogen atoms was much smaller than the size of Ni atoms.

The simulation was conducted using a large-scale atomic/molecular massively parallel simulator (LAMMPS). Equations of motion for the atoms were integrated numerically using the fourth-order Verlet method with the time step of 0.2 fs. The Nose–Hoover thermostat was used to control the system temperature. The periodic boundary conditions were applied in all directions, however, the simulation box is much bigger than the size of GF filled with nanocluster. The adaptive intermolecular reactive empirical bond order potential (AIREBO) [56] was used to describe the interatomic interactions between carbon atoms, including both covalent bonds in the basal plane of graphene and van der Waals interactions between GF and nanocluster. The simulation configurations were visualized by Visual Molecular Dynamics (VMD) Software [57].

### 2.1. Graphene with Nickel Nanocluster

Graphene flake with the nanocluster was exposed at 300 K and 1000 K for 20 ps to study the dynamics of nanocluster inside graphene flake. Previously [41], it was shown that the melting temperature of the Ni nanocluster was about 1300 K. Thus, in the present work, the highest considered temperature was 1000 K. At this temperature, Ni nanocluster was close to melting but not melted yet, even for the smallest nanocluster, which consisted of 21 atoms. Initially, Ni atoms were packed into the face-centered cubic lattice.

To describe the interatomic interaction between Ni–Ni and Ni–C, Morse interatomic potential was used with the parameters $D_{e}$ = 0.4205 eV, $R_{e}$ = 2.78 Å and β = 1.4199 1/Å for Ni–Ni [58]; and $D_{e}$ = 0.433 eV, $R_{e}$ = 2.316 Å, β = 3.244 1/Å for Ni–C. The parameters of the Morse potential for describing the interaction of nickel and carbon atoms were recently proposed using *ab-initio* simulation [59,60].

### 2.2. Graphene with Hydrogen Nanocluster

Graphene flake with hydrogen nanocluster was exposed at 77 K and 300 K for 20 ps. The temperature of 77 K was chosen since it was previously shown that better sorption of hydrogen can be found at 77 K [61]. The temperature of 300 K was chosen to study the dynamics at room temperature, however, it is known that this temperature is too high for hydrogen storage [46].

To describe the interatomic interaction between C and H, AIREBO interatomic potential was used.

It should be mentioned that the initially obtained structure contains atomic hydrogen, however, H atoms transform to H${}_{2}$ molecules and just several H atoms remain single (see Figure 1b). As it was shown, the lowest binding energy between graphene and H${}_{2}$ was observed when the distance between C and H${}_{2}$ in the range from 2.9 to 3.2 Å [61]. Hydrogen molecules can be bonded by van der Waals interaction when they move close enough to the side of GF. Single hydrogen atoms can be chemically adsorbed on graphene by covalent bonding.

## 3. Results

### 3.1. Graphene with Ni Nanocluster

At first, the evolution of the potential energy of the system during the crumpling process for graphene flake filled within spherical the Ni nanocluster was analyzed. In Figure 2, potential energy as the function of exposure time at 300 K (a) and 1000 K (b) for five types of nanoclusters was shown. It was found that the total potential energy of the system was saturated to a practically constant value at the end of the equilibration process, indicating that the system reached equilibrium and a stable state. All the changes in the energy curves correspond to some structural changes.

The longest time of stabilization at 300 K was found for Ni${}_{21}$ (5 ps). Curves for nanoclusters of close diameter are almost the same. Graphene flake with Ni${}_{38}$ and Ni${}_{47}$ reach the equilibrium state at about 4 ps of exposure, while the GF with Ni${}_{66}$ and Ni${}_{78}$ reach equilibrium state at 3.2 ps. The bigger the diameter of the nanocluster, the less the equilibration time. This can be explained by the mutual arrangement of the nanocluster and GF. For a small nanocluster, the distance *a* is two times higher than for the biggest one which means that the time required to attach the nanocluster by the graphene flake is longer. For nanocluster Ni${}_{78}$, 3 ps is enough for GF to fully cover the nanocluster, while for Ni${}_{21}$, not only coverage took place, but also the further crumpling of the flake with the changing of the round shape of nanocluster.

At higher temperature (1000 K), again a strong correlation between equilibration time and the size of the nanocluster was found. For GF with Ni${}_{21}$ nanocluster, the transformation was fast since the temperature was close to the melting point and the nanocluster can be easily destroyed. Temperature fluctuations facilitate the crumpling process and the appearance of new bonds between the edges of GF. For bigger nanoclusters, the temperature slightly affects the time of equilibration. Again, the same values of equilibration time were obtained for Ni${}_{38}$ and Ni${}_{47}$ (about 4 ps) and for Ni${}_{66}$ and Ni${}_{78}$ (about 3.2 ps). However, temperature decreases the total potential energy of the system.

In Figure 3, the process of crumpling for nanoclusters Ni${}_{21}$ (a), Ni${}_{47}$ (b) and Ni${}_{66}$ (c) was shown in details after exposure at 300 K (blue line) and 1000 K (red line). At the beginning of the crumpling process, GF startED to change its round shape, and the fcc crystalline order of Ni nanocluster WAs destroyed. Metal atoms are attracted by the graphene surface and tend to occupy equilibrium positions above the center of the carbon hexagon, which was also mentioned in [41]. Ni atoms WEre interacting with graphene hexagons by van der Waals forces. The edges of GF can be bonded during exposure.

At first, consider GF with Ni${}_{21}$ (see Figure 3a, lower line of snapshots). At 300 K, GF lost its round shape at about 0.5 ps, edges of GF move towards each other, and at *t* = 1.5 ps one covalent bond appeared between two edges. Nanocluster disturbed by the temperature and several atoms attach to the graphene plane (state II). This leads to the spreading of the Ni atoms over the graphene plane (state III). Simultaneously, such a small nanocluster allowed the graphene flake to easily bend and the structural unit transforms to the bi-layer graphene with Ni atoms spread inside (state IV). Several more covalent bonds appeared on the side edges of GF. At 300 K, not many covalent bonds between the edges of GF can appear, since the temperature is far from the melting temperature of graphene [62,63].

Increase in the exposure temperature to 1000 K facilitates the transformation. At 300 K, equilibration time is equal to 5 ps, while at 1000 K, it is equal to 3.5 ps. Subsquently, the structure came to a stable state and no further changes were observed despite slight thermal fluctuations. At 1000 K, more carbon atoms on the edges of GF found neighbors since the structure is disturbed by thermal fluctuations.

For structures with Ni${}_{47}$ and Ni${}_{66}$, the behavior is qualitatively close. An almost round Ni nanocluster inside the graphene flake was observed at the initial state at both 300 K and 1000 K (as can be seen in Figure 3b,c). The edges of GF can attach to each other with the formation of new covalent bonds. During exposure, graphene flake transforms into a capsule containing nickel nanocluster. The nickel nanocluster almost completely fills the graphene flake, in comparison with Ni${}_{21}$. As a result, it is difficult to deform such a structure. However, at 1000 K, the nanocluster is more disturbed and the stable state (state III, the top line of snapshots) reached at a lower equilibration time. The nickel nanocluster became more planar since GF is rigid and can bend the soft metal nanocluster. In the case of Ni${}_{47}$, GF transforms the “bag” for nickel, while Ni${}_{66}$ nanocluster is too big and edges on both sides cannot be attached. In the case of Ni${}_{66}$, GF just covers the Ni nanocluster as much is possible.

A graphene flake with Ni${}_{38}$ behaves in a similar way to Ni${}_{47}$, while GF with Ni${}_{78}$ behaves similarly to Ni${}_{66}$. Note that GF always tends to wrap the metal cluster.

One of the important applications of such structural elements is the fabrication of composites. In Figure 4, the initial structure (a, a’) was composed of graphene flake filled with Ni nanocluster Ni${}_{21}$ and Ni${}_{78}$ correspondingly. The initial structure is compressed at 1000 K to obtain graphene–nickel composite material (c, c’). Composite can be fabricated under hydrostatic compression at high temperatures [41,42,43,44].

### 3.2. Graphene with Hydrogen Nanocluster

In Figure 5, the potential energy of GF filled with hydrogen cluster during exposure at 77 K (a) and 300 K (b) is presented as the function of equilibration time. Similar to GF with the Ni nanocluster, five structural units were divided into three groups—GF with 21 hydrogen atoms; GF with 38 and 47 H atoms; and GF with 66 and 78 H atoms. Despite that the hydrogen atoms transform into hydrogen molecules, the initial number of atoms was also used, since the number of H${}_{2}$ molecules and single hydrogen atoms can change from one simulation run to another which depends on thermal fluctuations.

At 77 K (Figure 5a), the equilibrium state is the state when hydrogen atoms found their sits and GF change the cylinder shape to the one with minimal energy. At 300 K (Figure 5a), the equilibrium state is the state when all hydrogen atoms disrobed from the side of GF or even fully leave the cavity of GF. This would be discussed later together with the description of corresponding snapshots. The biggest drop of the energy value took place during the first picosecond and connected with the change of initial nonequilibrium configuration of the cluster and slightly with the change of shape of the GF.

At 77 K, the time of equilibration was the longest (3 ps) for hydrogen clusters consist of 77, 66, and 38 atoms, and the shortest (1.5 ps) is for 21 and 47 atoms. At that temperature, the size of the cluster plays quite an important role which is connected with the number of sites for hydrogen on the side of GF. All hydrogen molecules and atoms can easily find sites on graphene for an initial number of atoms less than 47, while for bigger clusters, the number of sites is not enough and some molecules and atoms will move outside GF or seek better sits near GF.

At 300 K, the main point is the process of dehydrogenation which is quite quick at such a high temperature, and for clusters with 38–78 atoms, about 1.5 ps is enough for all molecules to detach the side of GF and move outside. The long time of equilibration for cluster H${}_{21}$ can be explained simply: there are about nine molecules and three atoms and all the time spent for the slow movement of this hydrogen. A large number of atoms in the cluster pushes the sides of GF and opens it much faster.

To understand the dynamics of the hydrogen sorption/desorption, snapshots of the structural units altogether with their potential energy are presented in Figure 6 for three groups. If the hydrogen cluster is quite small (21 atoms), there are a lot of vacant places on the graphene plane to attach hydrogen. Only several atoms or molecules can move outside GF at a low temperature equal to 77 K. Most of the hydrogen molecules attached by van der Waals force to graphene and slightly moving along graphene flake.

At 300 K, hydrogen molecules and atoms have no chance to form even a weak bond to graphene. At 300 K, all hydrogen atoms move outside GF which is quite understandable. Numerous theoretical and experimental works confirm the sorption of molecular hydrogen on carbon nanostructures [50,64,65,66] in a specific temperature range (50–200 K). Here, such a high temperature was used to analyze how hydrogen will leave the cavity of GF. As it can be seen from Figure 6a, this process is very fast and at 4 ps, only two molecules stay inside GF (stage IV, upper line).

A large number of hydrogen atoms almost completely fills the cavity of the GF. At 77 K, hydrogen atoms are easily converted into hydrogen molecules and under the action of chemical attraction or van der Waals forces, attach to the walls of the graphene flake. Mainly, hydrogen, which is located in the center of the flake under the influence of temperature, tends to leave the graphene cavity. Therefore, in structures with 66 and 78 hydrogen atoms, the stabilization of the structure takes longer (3 ps), in contrast to small hydrogen clusters, such as 21 and 47 (1.5 ps).

For a bigger nanocluster, energy curves almost coincide for two temperatures, since hydrogen molecules at 77 K can find the places for sorption just after the first steps. In this case, hydrogen atoms are placed near graphene and link it as far as the van der Waals radius of the molecule reaches. Thus, the hydrogen adsorption capacities depend considerably on the initial size of the hydrogen cluster. The SSA of the GF should be big enough for a chosen number of H atoms and H${}_{2}$ molecules. Here, SSA is equal to 1153.9 m/g${}^{2}$, which is enough to settle down 21–38 H atoms, but not enough for a bigger number of H atoms.

In Figure 7, snapshots of GF with 78 hydrogen atoms during exposure at 77 K for 20 ps are presented. As it can be seen, some hydrogen molecules attach the opposite side of GF since there were no sites inside the cavity. During exposure, even at 77 K, GF opens and then closes again. If the exposure time would be increased to even 100 ps, GF will move like the wings of a flying bird with adsorbed hydrogen atoms. This state is equilibrium and can be preserved for a long time.

As well as for a metal nanocluster, the three-dimensional structure of crumpled graphene filled with hydrogen is presented in Figure 8. The corresponding structure was considered in [46,47], where the effect of hydrostatic compression on the possibility of hydrogen storage was studied. In Figure 8a, the initial structure of crumpled graphene was presented. As it can be seen, initially there are too many channels for hydrogen to move out of the structure. Thus, additional treatment was required to save hydrogen inside the pores of crumple graphene. In Figure 8b, the structure after 40% of hydrostatic compression was presented. In such a compressed structure, hydrogen can be stored much more effectively than in undeformed crumpled graphene [46,47]. It can be concluded that crumpled graphene is an effective storage media for hydrogen. However, a search for the improvement of the quantity of stored hydrogen should be found, for instance, in the introduction of metal atoms. From this point of view, understanding the interaction between hydrogen and metal nanoclusters and GF is of high importance.

## 4. Conclusions

Molecular dynamics simulation is used to study the dynamics of graphene flakes filled with the nanoclusters of two different types: metal and non-metal. The obtained results can shed the light on understanding the possibility of using crumpled graphene as a storage media from the point of single flake behavior.

It was found that cavities of crumpled graphene (the structure consists of crumpled graphene flakes) can be used as containers for metal nanoclusters—for instance, for Ni. A nanocluster consisting of 21 to 78 Ni atoms was considered. A graphene flake can easily cover nanocluster, however, the dynamics of the interaction strongly depend on the nanocluster size. Small nanoclusters can be easily bent by rigid graphene flake, while the biggest conserve the shape. Such a structure, composed of Ni nanoclusters and graphene flakes, can be further used to obtain composite material with improved mechanical properties.

The problem of hydrogen storage has been of high importance for decades and the application of carbon nanostructures from this point of view also looks promising. Crumpled graphene has a high specific surface area, light weight and a lot of pores and cavities which can be filled with hydrogen. It was shown that at 77 K, hydrogen molecules and atoms can be absorbed by both sides of graphene flakes. However, a single flake cannot store enough hydrogen for practical application. However, when graphene flakes composed in another structure and with additional treatment like hydrostatic compression [46,47], it can be successfully used for hydrogen storage and transportation.

## Figures and Tables

**Figure 1 materials-14-02098-f001:**
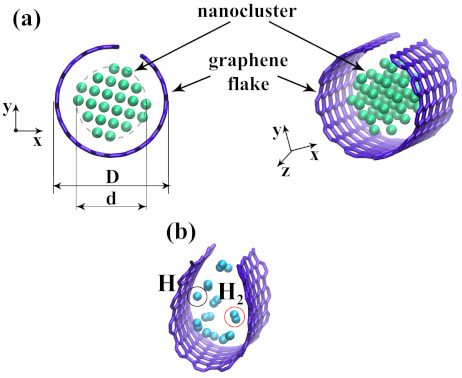
(**a**) Initial structure: graphene flake with the diameter *D* filled with nanocluster with the diameter *d*; (**b**) atomic and molecular hydrogen inside the graphene flake.

**Figure 2 materials-14-02098-f002:**
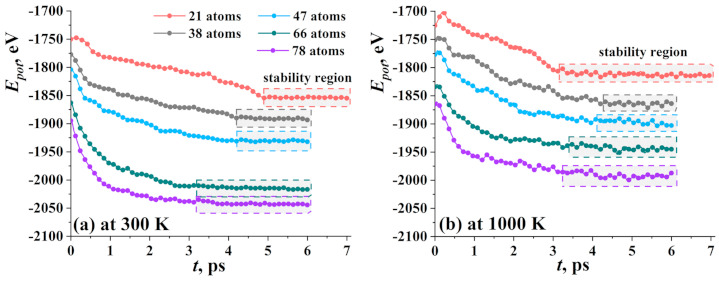
Potential energy as the function of exposure time at 300 K (**a**) and 1000 K (**b**) for five types of nanoclusters.

**Figure 3 materials-14-02098-f003:**
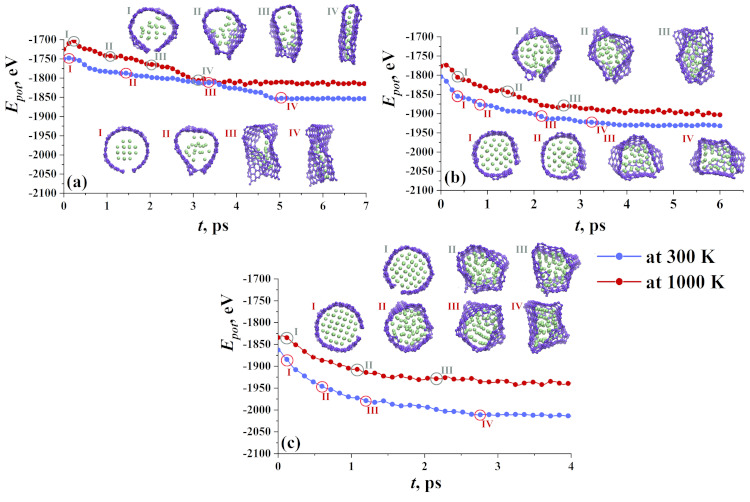
Potential energy as the function of exposure time for Ni${}_{21}$ (**a**), Ni${}_{47}$ (**b**) and Ni${}_{66}$ (**c**). Corresponding snapshots of the structure of crumpled graphene flake filled with nanocluster during exposure at 300 K (bottom line of snapshots) and 1000 K (upper line of snapshots). Carbon atoms are shown by violet and nickel atoms are shown by green.

**Figure 4 materials-14-02098-f004:**
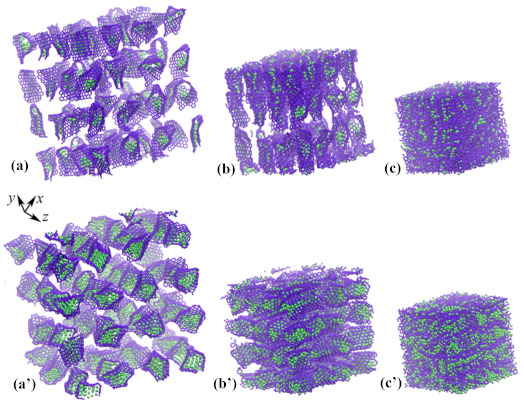
Snapshots of the structure of crumpled graphene filled with (**a**–**c**) Ni${}_{21}$ and (**a’**–**c’**) Ni${}_{78}$ nanocluster. Initial structure was obtained by annealing at 300 K, while composite was obtained by hydrostatic compression at 1000 K. **Ni** atoms are shown by purple and **C** atoms are shown by green color.

**Figure 5 materials-14-02098-f005:**
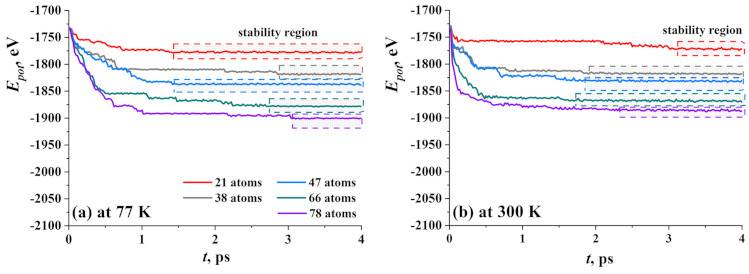
Potential energy as the function of exposure time at 77 K (**a**) and 300 K (**b**) for five types of hydrogen nanoclusters.

**Figure 6 materials-14-02098-f006:**
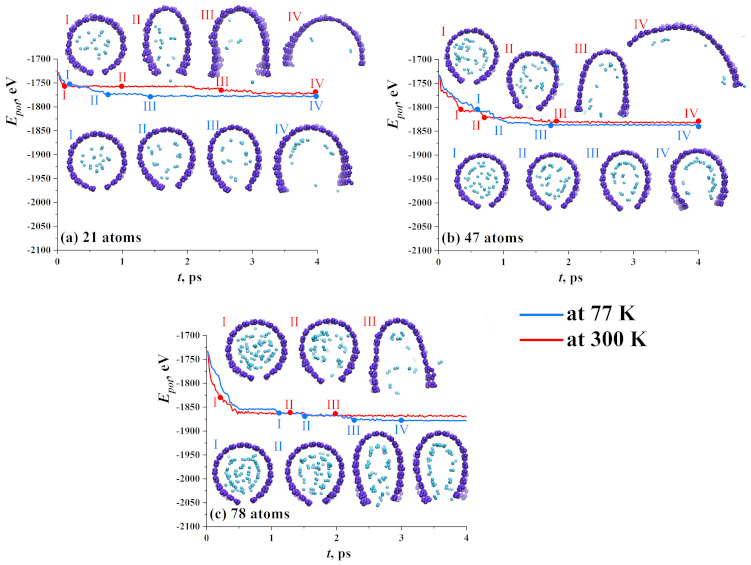
Potential energy as the function of exposure time for three types of hydrogen nanoclusters: (**a**) 21 atoms; (**b**) 47 atoms; and (**c**) 78 atoms. Corresponding snapshots of graphene flake filled with nanocluster during exposure at 77 K (bottom line of snapshots) and 300 K (upper line of snapshots). Carbon atoms are shown by violet and hydrogen atoms are shown by blue.

**Figure 7 materials-14-02098-f007:**
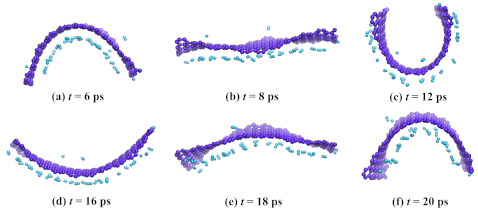
Snapshots of graphene flake with 78 hydrogen atoms during exposure at 77 K for 20 ps. Colors are as in Figure 6.

**Figure 8 materials-14-02098-f008:**
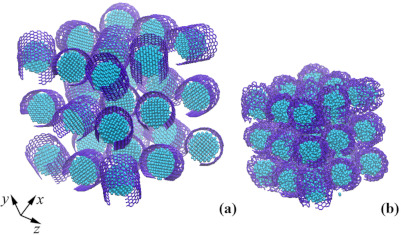
Snapshots of crumpled graphene filled with hydrogen atoms before (**a**) and after (**b**) hydrostatic compression. Colors are as in Figure 6.

**Table 1 materials-14-02098-t001:** The size of the structural elements.

*D*, Å	*N*	*d*, Å	*a*, Å
	21	6.2	4.25
	38	8.1	3.30
14.7	47	9.0	2.85
	66	9.4	2.65
	78	10.4	2.15

## Data Availability

Not applicable.

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
