# Peer review of "Crumpled Graphene-Storage Media for Hydrogen and Metal Nanoclusters"

_materials, 2021, doi:10.3390/ma14092098_

Round 1

Reviewer 1 Report

After reading the manuscript entitled "Crumpled graphene - storage media for hydrogen and metal nanoclusters " by Safina et al. I believe that the work is interesting and prepared correctly according to the art of scientific work. However, it requires a little enrichment.
My general remarks are :
- the organisation and writing of the paper are of good quality,
- the presentation of the adsorbent models and simulated systems is overall clear and easy to follow apart from some missing details,
- the discussion of the results is clear and most conclusions are sound and relevant. A few questionable assumptions should be revised, though,
- the paper opens up interesting perspectives for further improvement.
Hereunder are my detailed comments:
1.    Page 1, keywords. Please add “hydrogen”.
2.    Page 2, line 48. [?]? Please add the respective reference(s).
3.    Page 3, Fig. 1. Please use the same colours of Ni atoms and hydrogen atoms/molecules as in Figs. 3, 4, 6, and 7.
4.    Figs. 1, 3, 4, 6, and 7. What program was used to prepare these figures? VMD? Please add the respective reference(s)?
5.    Page 2, lines 53 and 59. Aims/goals? I do not fully understand the concepts of this work. The enhancement of the hydrogen storage?
6.    Page 3, Tab. 1. 9 ---> 9.0. In the case of “a” the values are given with too little accuracy, e.g. 4.2 ---> 4.25.
7.    Materials and methods. Why the authors did not examine the system GF/Ni/H(H2)? After all, the authors write in the introduction about carbon nanostructures doped with metals!
8.    Page 4, line 104. fig. 2 ---> Fig. 2.
9.    Fig. 2, y-axis. The same limits of the scale should be used - see Fig. 5. I would standardize it for the other figures.
10.    Fig. 3. I do not understand the choice of Ni66. Ni78 should be analysed as the extreme system!!!
11.    Fig. 4(a) and (a’). Can the systems shown in (a) and  (a’) exist as isolated?
12.    Figs. 2, 3, 5, and 6. Is potential energy analysis a good measure of achieving equilibrium? What parameter should be the measure of reaching the state of equilibrium?
13.    In Fig. 4 GF/Ni systems were studied. What abot GF/H(H2)?

Author Response

Reply to the comments of the Reviewer 1

We are very grateful for the great efforts and valuable suggestions made by the Reviewer. The detailed response to the comments of the Reviewers is provided below together with the description of the changes applied to the manuscript. We note that all changes to the manuscript are highlighted in red for the convenience of the reviewer.

Comment 1

Page 1, keywords. Please add “hydrogen”.

Reply:

We appreciate the comment and thank the referee for this advice. Keyword “hydrogen” is added.

Comment 2

Page 2, line 48. [?]? Please add the respective reference(s).

Reply:

We thank the referee for such a careful reading. Reference was added.

Comment 3

Page 3, Fig. 1. Please use the same colours of Ni atoms and hydrogen atoms/molecules as in Figs. 3, 4, 6, and 7.

Reply:

We appreciate the comment. Colors in Fig. 1 were changed.

Comment 4

Figs. 1, 3, 4, 6, and 7. What program was used to prepare these figures? VMD? Please add the respective reference(s)?

Reply:

We appreciate the comment. VMD software was used to visualize the structures in Figs. 1, 3, 4, 6, and 7. Reference was added.

Comment 5

Page 2, lines 53 and 59. Aims/goals? I do not fully understand the concepts of this work. The enhancement of the hydrogen storage?

Reply:

We appreciate the comment. The aim of this work is to study the interaction of a carbon structure with metallic and hydrogen clusters for their further storage in the pores of crumpled graphene. This idea is based on the intention to find a better structure for hydrogen storage. Various carbon structures are shown to be very promising for hydrogen storage, while still there is no such structure that can be applied in practice. Crumpled graphene is a structure with a considerable number of pores that can be filled with something else, some other atoms. But before the study of storage in three-dimensional crumpled graphene, it is very important to understand the dynamics of interaction between hydrogen and graphene flake. In the present work, we show the idea to use crumpled graphene flake for storage of different atoms using Ni and hydrogen clusters as the example. Corresponding explanations are added to the text.

Comment 6

Page 3, Tab. 1. 9 ---> 9.0. In the case of “a” the values are given with too little accuracy, e.g. 4.2 ---> 4.25.

Reply:

We appreciate the comment. Figures in the Tab. 1 are changed.

Comment 7

Materials and methods. Why the authors did not examine the system GF/Ni/H(H2)? After all, the authors write in the introduction about carbon nanostructures doped with metals!

Reply:

We appreciate the comment. The reviewer is totally right about our future goal to study the effect of metal atoms on the increase of hydrogen storage capacity in crumpled graphene. And this work is already started by our group. However, there are some simulation issues that should be examined before such work can be published. Now we can understand in detail the interaction between graphene flake and separately nickel and hydrogen atoms. Also, we have considered the behavior of 3D crumpled graphene with different fillers. For the system with both atoms different potentials should be checked also, and different external effects. This work is in progress. Thus, in the frames of the present work GF/Ni/H(H2) system is not presented.

Comment 8

Page 4, line 104. fig. 2 ---> Fig. 2.

Reply:

We appreciate the reviewer for such a careful reading. Misprint is fixed.

Comment 9

Fig. 2, y-axis. The same limits of the scale should be used - see Fig. 5. I would standardize it for the other figures.

Reply:

We appreciate the comment. We have changed the scale in Figs. 2 and 5. We also fixed this problem for Figs. 3 and 6.

Comment 10

Fig. 3. I do not understand the choice of Ni66. Ni78 should be analyzed as the extreme system!!!

Reply:

We appreciate the comment. The reviewer is totally right, the case for Ni78 is extreme. The difference between the diameters of 78 and 66 nanoclusters is small, but the point is that Ni78 stays too close to the sides of graphene flake which can negatively affect the interaction in the model. Our goal was to find the limitation in the ratio between the size of the nanocluster and graphene flake and we have studied Ni78 as the biggest possible. Results for Ni78 and Ni66 are close. However, to exclude the negative effect of too big size for this flake, we choose Ni66 since for further work we decided not to use Ni78. And when we will work on the continuation of this work we could have a possibility to base on the results of the present manuscript. Corresponding explanation is added to the text.

Comment 11

Fig. 4(a) and (a’). Can the systems shown in (a) and (a’) exist as isolated?

Reply:

We appreciate the comment. In fact, such a system cannot be obtained in real experiments and thus we cannot say it can exist isolated. In our model, we use such an idealized initial structure for simplicity and do not consider it as a system for property investigation. In MD it can exist but physically flakes too close to be interconnected by van-der-Waals. We apply hydrostatic compression until the empty spaces are removed and after that obtain a much more realistic case of crumpled graphene. The structure presented in Fig 4(b) and (b’) definitely can exist. Commonly, we start to analyze the properties from structure (b).

Comment 12

Figs. 2, 3, 5, and 6. Is potential energy analysis a good measure of achieving equilibrium? What parameter should be the measure of reaching the state of equilibrium?

Reply:

We appreciate the comment. In the present work, our idea was to find the structural state of graphene flake filled with other atoms with minimal potential energy after simple relaxation in molecular dynamics. We also take into consideration the equilibrium sites of Ni and hydrogen atoms, which are the same as in other simulation works. Graphene flakes and nanoclusters were also equilibrated and studied at different conditions separately. For the goals of the present work, such consideration is enough.

Comment 13

In Fig. 4 GF/Ni systems were studied. What abot GF/H(H2)?

Reply:

We appreciate the comment. Fig. 8 is added to show the 3D crumpled graphene filled with hydrogen. This structure was studied in our previous works and it was shown that hydrostatic compression is an effective method to save hydrogen inside the pores of crumpled graphene. Additional descriptions are added to the text.

Reviewer 2 Report

General comments Manuscript needs revisions. Specific comments Manuscript is filled with self-citations which are irrelevant in most cases. Ex: Authors say "new composite materials can also be obtained from Ni nanoclusters and crumpled graphene [29–32] or based on metal matrix and Ni coated-graphene" and self cite papers published in 2019 and 2020. Whereas growth of graphene on Ni is known way back in 2009. Ex: 10.1021/nl902515k, 10.1002/adma.200803016, 10.1007/s12274-009-9059-y etc. Authors are strongly advised to read and cite this paper titled "Graphene–nickel interfaces: a review" 10.1039/C3NR05279F. Secondly authors never justified why they chose Nickel when more effective metals like palladium are capable of storing more hydrogen. In 2. Materials and Methods: Authors say "One graphene flake is a small CNT(11,11) 1.3 nm long with two atomic raws deleted along the axis of the nanotube." But rather than calling this structure as 'crumpled graphene' it can be called as 'graphene nano-ribbon'.

Author Response

We are very grateful for the great efforts and valuable suggestions made by the Reviewer. We have revised the manuscript accordingly. The detailed response to the comments of the Reviewer is provided below together with the description of the changes applied to the manuscript. We note that all changes to the manuscript are highlighted in red for the convenience of the reviewer.

Reply to the comments of the Reviewer 2

Comment 1

Manuscript is filled with self-citations which are irrelevant in most cases. Ex: Authors say "new composite materials can also be obtained from Ni nanoclusters and crumpled graphene [29–32] or based on metal matrix and Ni coated-graphene" and self cite papers published in 2019 and 2020. Whereas growth of graphene on Ni is known way back in 2009. Ex: 10.1021/nl902515k, 10.1002/adma.200803016, 10.1007/s12274-009-9059-y etc. Authors are strongly advised to read and cite this paper titled "Graphene–nickel interfaces: a review" 10.1039/C3NR05279F.

Reply:

We thank the referee for this valuable comment since literature review improvement considerably increases the readability of the manuscript and the soundness of the results. Ni-graphene interfaces have been studied for a decade now and we have found a lot of interesting papers during working on the literature review. For this paper, we decided to pay more attention to the aspects of graphene crumpling more than to the explanation of the choice of metal (Nickel, for example) or to the review of Ni-graphene interface. However, we strongly agree with the referee that it is very important to show that the study of Ni-graphene coatings has a long history. Works mentioned by the reviewer, especially "Graphene–nickel interfaces: a review", help us to compensate for this deficiency. We didn’t meet this review previously and find it important for understanding the basics of the interaction of Ni and graphene. Additional text is added to the manuscript.

However, all the self-citations are used in the context since authors are working with 3D carbon nanostructures, especially pure crumpled graphene for a long time. In the reference list, there are no citations far from the subject of the present manuscript. Our works [2,5,13,21] are dedicated to the study of different aspects of crumpled graphene investigation. We use works [29-32] to show our experience in this field and no other works, studying exactly crumpled graphene filled with Ni nanoparticles, can be found in the literature. However, there are a lot of works on the other types of composites base on graphene and Ni, for example, layered and we have added this works to the literature review. Works [34,35] shows our experience in hydrogen storage in crumpled graphene.

Comment 2

Secondly authors never justified why they chose Nickel when more effective metals like palladium are capable of storing more hydrogen.

Reply:

We appreciate the comment. At first, a short explanation of the choice of metal is added to the text. And second, we wish to explain this to the esteemed reviewer in this letter wider. In fact, this work was done in the frames of the large work on the study of metal-graphene composites. We are planning to study a variety of metals – Ni, Al, Pd, Au and Cu in different forms – as the matrix for graphene flakes and as the fillers for crumpled graphene and to compare mechanical properties. And one of the main ideas is the application of crumpled graphene for storage (for metal or non-metal atoms). This work is one of the first steps of such a complex study: we have started from Ni since we got very good Morse parameters for Ni (obtained by ab-initio) and, of course, since it quite important metal for different applications. To understand the simulation methodic, the interaction between structural elements, and the process of obtaining the composite we need to start from a quite simple and understandable approach. So we choose Ni as well-studied material especially in the combination with graphene, for which we have well-described potential, and then we studied the interaction of one particle and one graphene flake. This comment of the reviewer is very clear for us and the study of graphene-Pd system is one of our next goals. But in the frames of this work, we consider just Ni case.

Comment 3

In 2. Materials and Methods: Authors say "One graphene flake is a small CNT(11,11) 1.3 nm long with two atomic raws deleted along the axis of the nanotube." But rather than calling this structure as 'crumpled graphene' it can be called as 'graphene nano-ribbon'.

Reply:

We appreciate the comment. Here we are considering very short CNT, while graphene nano-ribbon is an elongated 1D structure with the nanoribbon length many times bigger than lateral size. The length of the side of the flake (along z) is 1.3 nm, along y is about two times bigger, along x it is one-atom-thick. In fact, our graphene flakes closer to 0D structure with three very low dimensions. Thus, we chose the term “flake” and since it is crumpled during treatment we call it “crumpled graphene flake”. We have added this explanation to the text also.

Round 2

Reviewer 1 Report

The authors have made a substantial improvement for this article. The manuscript can be accepted for publishment in the present form.

Reviewer 2 Report

Authors have made suitable changes/modifications in revised manuscript. Accept